# Behaviour-based movement cut-off points in 3-year old children comparing wrist- with hip-worn actigraphs MW8 and GT3X

Daniel Jansson[1,2], Rikard Westlander[3], Jonas Sandlund[4], Christina E. West[3], Magnus Domellöf[3]*, Katharina Wulff [5,6]*

1 Department of Community Medicine and Rehabilitation, Section of Sports Medicine, Umeå University, Umeå, Sweden, 2 Umeå School of Sport Sciences, Umeå University, Umeå, Sweden, 3 Department of Clinical Sciences, Pediatrics, Umeå University, Umeå, Sweden, 4 Department of Community Medicine and Rehabilitation, Section of Physiotherapy, Umeå University, Umeå, Sweden, 5 Department of Molecular Biology, Umeå University, Umeå, Sweden, 6 Wallenberg Centre for Molecular Medicine (WCMM), Umeå University, Umeå, Sweden

* katharina.wulff@umu.se (KW); magnus.domellof@umu.se (MD)

## Abstract

### Background

Behaviour-based physical intensity evaluation requires rigorous calibration before application in long-term recordings of children's sleep/activity patterns. This study aimed at (i) calibrating activity counts of motor behaviour measured simultaneously with Motion-Watch 8 (MW8) and ActiGraph (GT3X) in 3-year-old children, (ii) documenting movement intensities in 30s-epochs at wrist/hip positions, and (iii) evaluating the accuracy of cut-off agreements between different behavioural activities.

### Methods

Thirty 3-year-old children of the NorthPop cohort performed six directed behavioural activities individually, each for 8–10 minutes while wearing two pairs of devices at hip and wrist position. These naturally-occurring behaviours were aligned to movement intensities from 'motionless' (watching cartoons) and 'sedentary' (recumbent story listening, sit and handcraft) to 'light activity' (floor play with toys), 'moderate activity' (engaging in a brisk walk) and 'vigorous activity (a sprinting game). Time-keeping was ensured using direct observation by an observer. Receiver-Operating-Curve classification was applied to determine activity thresholds and to assign two composite movement classes.

### Results

Activity counts of MW8 and GT3X pairs of wrist-worn (rho = 0.94) and hip-worn (rho = 0.90) devices correlated significantly (p < 0.001). Activity counts at hip position were significantly lower compared to those at the wrist position (p < 0.001), irrespective of device type. Sprinting, floorball/walk and floorplay assigned as 'physically *mobile'* classes achieved outstanding accuracy (AUC > 0.9) and two sedentary and a motionless activities assigned into 'physically *stationary'* classes achieved excellent accuracy (AUC > 0.8).

**Data availability statement:** Sources for raw data and manuals: RAW DATA EXCEL SPREADSHEET 10.6084/m9.figshare.27896760 ACTIGRAPHY OPERATIONAL MANUAL https://www.katlab.org/wp-content/uploads/2023/10/ACTIGRAPHY-OPERATIONAL-MANUAL-Nordic-Daylight-Research-Programme-2023.pdf

**Funding:** We thank for the financial support from the Swedish Research Council (Vetenskapsrådet grant number 2019-01005 to MD), Region Västerbotten (ALF research infrastructure grant) and Umeå University research infrastructure grants to MD and CW, as well as a Särskild satsning grant from the Wallenberg centrum för molekylär medicin (WCMM, proj no: FS 2.1.6-849-209) to KW. The work of KW was partially supported by the Knut and Alice Wallenberg Foundation. The work of DJ was funded by institutional funds from the department of community medicine and rehabilitation, Umeå University. None of the funders played any role in the study design, data collection and analysis, decision to publish, or preparation of the manuscript.

**Competing interests:** The authors have declared that no competing interests exist.

## Conclusion

This calibration provides useful cut-offs for physical activity levels of preschool children. Contextual information of behaviour is advantageous over intensity classifications only, because interventions will focus on behaviour-allocated time to reduce a sedentary lifestyle. Our comparative calibration is one step forward to behaviour-based movement guidelines for 3-year-old children.

## Introduction

Movement-related assessments of habitual activities, such as physical activity (PA) levels or sleep/circadian timing, have typically taken a segregated approach in fields such as epidemiology, sports medicine, rehabilitation or chronobiology [1–3]. Historically, terminology also developed independently, with 'actimeter/actigraph' used in sleep/chronobiology and 'accelerometer' in sports medicine (S1 Table). Similarly, evidence regarding the combination of movement behaviours over a 24-h period using compositional analyses [4] is uncommon but growing [5] with studies emerging [6–8]. The Commission on Ending Childhood Obesity recognised the importance of interaction among PA, sedentary behaviour and adequate sleep on the child's well-being [9]. Canadian and Australian 24-hour movement guidelines were developed [4,10], followed by World Health Organisation (WHO) guidelines on the amount of *time in a 24-hour day* particularly for children under five years of age [1]. Global recommendations state that children 3–4 years old should spend at least 180 minutes a day in a variety of physical activities at any intensity, of which at least 60 minutes is moderate- to vigorous-intense PA [1]. This PA recommendation is considered strong, albeit of very low-quality evidence [1].

### Challenges for compositional analysis

There are a number of calibration studies for wearable movement sensors using short epochs (5–15s) in preschool children [11–21] (S1 Table). Objective measurements of physical movements with wearables have proven feasible and valid for estimating sedentary time during waking hours in young children in the field [22–24]. However large-scale cohort recordings capture behaviour over many days and require the distinction of motionless alert from daytime sleep, which necessitates devices to be compared and calibrated for their accuracy in measuring PA and derived sleep [25,26]. Every device type needs to be calibrated for typical behaviours of a given population [27], and epoch as well as commonly used body sites, mainly the wrist and hip [28,29]. The challenges with accelerometer calibration, dividing-up absolute and relative intensities, have been competently outlined by Arvidsson et al. 2019 [30] and consensus recently discussed by Migueles et al. 2022 [31].

In brief, the standard placement of devices for PA alone has been at the hip by fitting it on an elastic belt around the waist [32], while devices developed for sleep/circadian rhythm research have typically been attached to the non-dominant wrist [33,34] (S1 Table). Accordingly, the devices' sensors and firmwares operate differently according to their purpose: Measurements for assessing sleep/circadian patterns last over complete 24h cycles and longer epochs (30 to 60s) for several weeks with sensors and firmware adjusted for wrist movements [35]; while sensors used for PA measurements are usually implemented for high-resolution (seconds), short-term (min-hours) measurements on the hip-position to determine energy expenditure [36–38]. Ramification are intricate, including systematically different activity levels from different brands, because their raw accelerometer outputs are unequal [39] or when using same brand devices, they classify sedentary behaviour and PA accurately under

laboratory settings but not under free-living conditions [40]. Scaling factors implemented into algorithms may overcome inherent discrepancies, provided that the research communities share data collections and meta-data from as many models as possible [2].

On the nature of compliance, wrist position was found to achieve higher compliance than hip position [41]. Therefore, preschool children are recommended to attach the device to the *non-dominant* wrist [42,43].

### Rational of the study

The aforementioned recommendations were based on systematic reviews with an overall 'moderate' certainty of the evidence, stating little research was available to inform about specific aspects such as dose-response studies on the type, duration, intensity and context of individual behaviours [4]. In turn, uncertainty was reported in how light PA and moderate-to-vigorous intensity PA is best defined for young children. Taken together, the guidelines recommend durations and PA quantities for several age ranges without accounting for different detection methods and different body positions to track behavioural situations. These gaps can be closed by: (i) comparative assessments of different devices; (ii) worn simultaneously at different body positions; (iii) during age-appropriate behaviours accounting for different PA intensities; (iv) in children at narrower age ranges.

However, many devices are on the market and researchers are confronted with analytical challenges in face of the technical diversity. The appropriateness in filter settings, algorithms, recording modes, size and position for the combination of PA and sleep are under scrutiny [44]. Furthermore, various research-grade data loggers capture light exposure levels due to its effect on circadian/sleep behaviour [45] in addition to body movements [46], at the expense of recording period and size. Two of those have been validated for sleep/circadian quantities and PA in adults and children: MotionWatch 8 (MW8, Camntech LTd, UK) and ActiGraph GT3X (theactigraph, FL, US). Both use different algorithms and output modes [47–49]. The MW8 is primarily used in longitudinal sleep/circadian studies, which prioritises a 30s epoch as a result of the validation with polysomnography to derive sleep parameters from movement patterns [50] and to allow longer recording periods. The GT3X uses the Cole-Kripke [51] and Sadeh et al. [52] algorithms as their standard sleep algorithms.

### The present study

The present study was specifically designed to address some of the aforementioned gaps by measuring PA intensities of different behaviours simultaneously at two body positions (non-dominant wrist and hip) with two models (MW8 and GT3X) using the same epoch length of 30s in children within a narrow age range (3.5 years). Both device types were expected to produce systematically different intensities for the same behaviour. It was important to investigate, how much of a difference it made for time spent '*physically mobile*' (body in motion) from '*physically stationary*' (motionless/sedentary activities, excluding naps/sleep). The two devices and positions were also necessary, because each type has generally favoured only one of the positions. Our design made a comparison of the distribution of intensity levels within and between device types across positions possible. We first piloted various activities and then chosen six different behaviours to calculate behaviour-based activities for which we determined cut-off boundaries and their accuracies. This design goes beyond general calibration and insights were anticipated in the accuracy of discriminating intensity classes with a 30s eoch length and how to interpret scores between positions and devices. Finally, we envisage to assign behavioural classes according to cut-off intensity boundaries in our prospective, population-based NorthPop birth cohort (https://www.katlab.org/ [under

'people'], www.northpop.se), compare their allocated time with movement targets for pre-school children and relate to health indicators.

## Materials and methods

### Participants

In total, 30 children were recruited from the ongoing prospective, population-based North-Pop cohort. Three-year-old, typically developing children were included. Children with any chronic disease or weight outside the normative range (±2 standard deviations) using a Swedish growth reference [53] were not included to minimise confounding variables, which could potentially skew the calibration of activity counts. This was decided to ensure that the cut-off points we develop are generalisable to the broader population of typically developing 3-year-old children.

Children in this study were part of a larger actigraphic project within the original North-Pop cohort (Dnr 2014/224-31), so all children had experience wearing actigraphs during free-living conditions. Therefore, all participants were already familiar with the procedure, and adding more devices was well-tolerated. Further, all legal guardians received written and oral information about the study and signed a written consent prior to the data collection. In addition to legal guardians' consent, we explained the procedures to the children in simple terms and demonstrated where the monitors would be placed using a teddy bear to ensure their understanding and comfort. This calibration study was conducted in agreement with the declaration of Helsinki and approved by the regional ethical review board (Dnr 2020/01254).

### Procedures

All measurements were conducted between October 2020 and January 2021. Each parent-child pair was studied separately on one test occasion in the E-health laboratory at Umeå University. The E-health laboratory was designed to mimic a small apartment to simulate daily living. We collected the data during daytime between 9.00 to 12.00 or 13.00 to 16.00. Basic anthropometric measurements were collected for each child. Height was measured to the nearest 0.1 cm using a folding rule and body mass to the nearest 0.1 kg using an electronic scale. A pilot evaluation of the feasibility of various activities was carried out before commencing the data collection. A final study protocol was developed to monitor three stationary behaviors and three physically demanding behaviors typical for children to engage in at this age under free-living conditions. The six behaviours represent different movement patterns and intensity levels: Physically *mobile* levels endorsed physically 'vigorous', 'moderate' and 'light' activities, and physically *stationary* levels endorsed 'sedentary crafts', 'recumbent listening' and 'sedentary screen time' (Fig 1 with explanatory table).

Sedentary screen time involved watching a cartoon on a large screen TV with their parent and recumbent listening involved a parent reading their child's favourite story. The 'sedentary crafts' activity involved the child sitting at a table, drawing or placing stickers in a sticker book. The three active behaviours were planned to be progressively more intense: Playing on the floor with toys represented as light activity (defined as little increase in heart rate, can talk while playing);, moderate activity (defined as increased heart rate and breathing rate) was represented by walking or playing floorball, and vigorous activity (defined as significantly high heart rate and breathing hard and fast) represented by competitive sprinting games. Here the children were asked to carry balls from a box on one side of a long corridor to another box at the other end as quickly as possible. They had short breaks to take their breath at each side and they did not run continuously fast for 10 min. Each activity was performed for approximately 10 minutes, with a minimum of 3 minutes rest between activities. If the activity

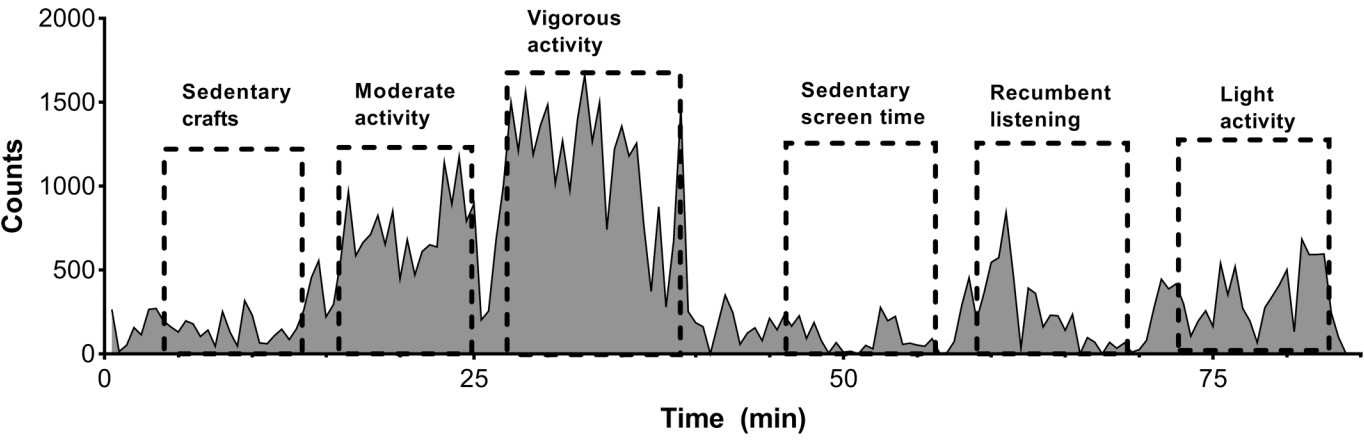

| Intensity | Activities | Description |
|---|---|---|
| Vigorous activity | Sprinting games | Competitive sprinting games against parent/test leader aimed at moving softballs between stations on each side of a corridor |
| Moderate activity | Walking/floorball | A moderate paced walk indoors or playing miniature floorball |
| Light activity | Playing on the floor | Playing actively on the floor with toys like cars, trains, or building blocks |
| Sedentary crafts | Arts and crafts | Sitting on the couch and drawing with pencils or playing with stickers |
| Recumbent listening | Listening (visual +auditory) | Lying quietly in a supine position, primarily listening to the parent reading their child´s favourite book, while it also looks at the illustrations |
| Sedentary screen time | Watching (visual and auditory) | Sitting quietly with parent primarily watching cartoons unfamiliar to the child on a large TV screen |

**Fig 1. Top: Representative example of an area graph derived from wrist activity patterns of a child performing six investigator-observed behaviours (dashed boxes) shown for a MotionWatch 8 device attached to the non-dominant wrist.** The activity intensity was transformed into count values recorded every 30 seconds (epochs). Each type of behaviour was observed by stop-watch to last between 8 to 10 minutes. Bottom: Explanatory table describing the six behavioural activities at the top.

was less than 8 minutes, we repeated that specific activity until it was within the time limits (8–10 min). The activities could be executed in any order depending on the child's preferences, whilst a relaxation break with a snack was offered after about half way through the activities. After the vigorous activity, the child had a little longer break (≈10 min) to minimize the influence of fatigue on the following activity. Two research team members were skilled to explain the study to the child and their parent, to handle the technical devices and software, and trained in momentary time sampling, which is here termed 'directly observed'. The 'direct observation protocol' entails one person acting as a 'bystander' watching (observing) each child's activities from a 'bird's perspective' and documenting each activity's start and stop with a stopwatch. This enabled the second person to engage with the child without being distracted by checking the time. The documented start and stop times were essential for extracting the sequences of activities from the times series. Children were fitted with four movement sensors (MotionWatch 8 ([MW8], CamNtech Ltd., Cambridge UK), ActiGraph [GT3X], ActiGraph, LLC, Fort Walton Beach, FL) and a heart-rate monitor (Actiheart 5, CamNtech), whose data were not part of the current analyses. A pair of MW8 and GT3X were attached to the non-dominant wrist and another pair to the hip. The hip-worn sensors were mounted to an elastic belt so that the black circle of the GT3X and the label of the MW8 were both facing downwards. The wrist-worn sensors were mounted to the same strap pointing towards the hand, following our standard operational procedures. The children wore the belt around the

waist, placed 1–2 cm beneath the umbilicus. The same four devices were used in all children during all activities to minimise variability between sensors. This set-up enabled comparisons between different types of sensors in the same position (wrist-wrist versus hip-hip) as well as within and between sensors across different positions (wrist versus hip). Missing data were rare and when discovered, participant would be replaced until we had complete datasets. However, for one participant we missed data for the MW8 at hip position.

Data from MW8 and GT3X were captured in 30-second epochs. The GT3X is a lightweight (27 g), relatively obtrusive (3.8 × 3.7 × 1.8 cm) device with a rechargeable battery. Batteries were charged before each trial to avoid missing data. The GT3X collects motion data on three axes (x, y, z) and is designed to record accelerations ranging from 0.05 to 2.5g. The sampling frequency of 30 Hz and "triaxial mode" was used for both GT3X devices and the "vector magnitude (VM)" for the analyses. The MW8 is battery-powered and weighs 9.1g. It is of relatively unobtrusive dimensions (3.6 × 2.82 × 0.94 cm), which is an advantage for long-term wear in younger children (e.g., 3-year-olds). Its continuous operation has the option of single-axis or tri-axial recording mode (producing a vector magnitude count per epoch). The MW8 has a built-in light sensor and a time-stamp marker button. The MW8 sensing ranges between 0.01 and 8 g, and the minimum 'not moving' threshold is 0.1 g. Data are sampled at 50 Hz and bandwidth-filtered between 3 Hz and 11 Hz. An 'activity count' (unitless) is derived from the highest of the 50 samples/second, and these values are accumulated over the length of an epoch (30 seconds) [54]. Both MW8 devices were set up to record in single-axis 'Motion-Watch Mode 1' and not tri-axial mode, because this mode has been validated against polysom-nography. It also allows longer recording periods, as implemented in the ongoing collection of sleep-wake and light exposure data in the Northpop cohort. After data collection, the raw time series data were downloaded using Motionware software and exported into an Excel spread-sheet. The start and stop times of the behavioural observation protocol were matched to the 8 to 10 minutes-long periods of raw time series data in Excel. Periods of the same behaviour across the children were successively concatenated, separately for each position and device. The total duration per behaviour was calculated and compared between the six behaviours to ensure similar data length distribution (S3 Fig).

## Statistical analysis

All statistical analyses were performed using the SPSS statistical package (SPSS, v. 27, Chicago, IL). Anthropometrical data at baseline were visually inspected using histograms, kurtosis, and skewness. Data were considered normally distributed and compared between males and females using a standard Student T-test. Activity count data of different intensities were not normally distributed and therefore the Spearman's-Rank-Order correlation was used to determine the monotonic relationship between the activity measured with the wrist-worn and hip-worn devices GT3X and MW8. To approximate the similarity in the shape of the data distribution between algorithms, the data of wrist-worn devices were analyzed with linear regression. The data of hip-worn devices showed a non-linear relationship, therefore poly-nomial nonlinear regression models were applied. Due to the skewed distribution of activity counts were plotted as boxplots in a log scale to visualise their proportional overlap between adjacent behavioural activities.

Receiver Operator Characteristics (ROC) curves were used to determine cut-off points (intensity thresholds) from activity counts of pre-defined, observer-based behaviours [55,56]. ROC curves have previously been used to determine activity count thresholds in children between 4–8 years for Actigraph and MW8 models separately [12,13,49]. Here we used both brands simultaneously and applied ROC curves to examine the classification accu-racy across brands and positions. We used a binarised approach in ROC curve classification

acknowledging the activity counts to be of descending order from vigorous PA> moderate PA> light PA> sedentary> motionless alert. Classification can be performed in two ways: pair-wise 'One-vs-One' (Fig 2a) or one-group/composite against all other 'One-vs-Rest' (Fig 2b). Although we report One-vs-One details in Supplementary Information (S2 Table), here we focus on the One-vs-Rest (OvR) scheme [57]. We collapsed counts of certain adjacent behavioural activities (and assumed as one) and compared those against all other collapsed behavioural activities (Fig 2b, see ROC curves in Supporting information S3–S7 Figs). The behavioral activities were assigned with intensity class labels: the most vigorous activity 'Sprinting' was assigned to 'vigorous PA' (VPA); 'Sprinting' and 'Floor ball' combined were assigned to 'moderate-vigorous PA' (MVPA); 'Sprinting', 'Floor ball' and 'Play on floor' combined were assigned to 'light-moderate-vigorous PA' (LMVPA); and all these together were labelled '*mobile* PA' class. They were set against all physically '*stationary*' behaviours, thereby dividing '*mobile PA*' from '*stationary PA*'. *Stationary* PA included 'sedentary crafts', 'recumbent listening' and 'sedentary screentime'. We merged 'sedentary crafts' and 'recumbent listening' behaviours into 'sedentary PA' (SED) on the basis of their considerable overlap in movement patterns. The most immobile behaviour 'sedentary screentime' was assigned to 'motionless alert' (MOA). We deliberly split off MOA from SED to better understand their closeness to quiet awake and sleep. We did not run a separate ROC-AUC for the SED, but report the range between lower boundary of LMVPA intensity and the upper boundary of MOA intensity.

To determine the best possible compromise between sensitivity and specificity for each cut-off, we applied the Youden Index (*J*), which is determined by calculating the sum of sensitivity and specificity minus one [58]. The cut-offs (synonym for intensity thresholds) were derived from the raw data distribution's highest total sensitivity and specificity ratio. All values are reported as mean ± standard deviation (SD). The Area Under the Curve (AUC) quantified how well the ROC curve performed at classifying data. Values for AUC ranges from 0 to 1. It is used as a measure of accuracy for the ROC curve in discriminating classes, here in

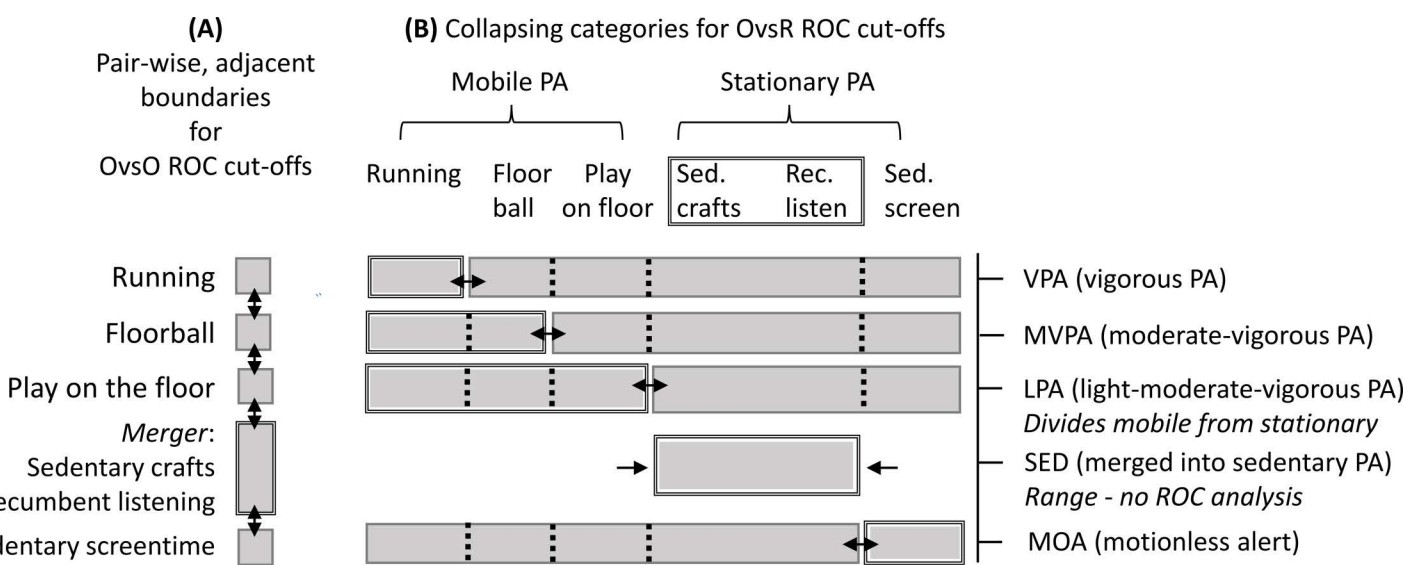

**Fig 2. Schematic description of ROC analysis procedures: A) One versus one (OvsO) and B) one versus the rest (OvsR) in order of descending intensities.** Double-sided arrows indicate the cut between categories for each ROC analysis. In OvsR, two sedentary categories (Sedentary craft + Recumbent listening) were merged into SED (for which ROC analysis were done only in OvsO, see S2 Table). Moreover, SED and MOA were collapsed into an composite '*stationary* PA' class and tested against the composite '*mobile* PA'.

classifying count cut-offs from pre-defined, observed behavioural activities. ROC-AUC values above 0.90 are considered outstanding, 0.80–0.89 excellent, 0.70–0.79 acceptable (fair), less than 0.70 are considered poor, and an AUC as low as 0.5 indicates no greater predictive ability than by random guessing [59]. A significance level of $p < 0.05$ was considered significant.

## Results

Anthropometrical data for all participants are presented in Table 1. The study included 30 participants, 20 males and 10 females. There was no significant difference in age ($p = 0.69$), height ($p = 0.898$) or waist circumferences ($p = 0.052$) between males and females. The males were heavier than females ($p = 0.023$), which was also reflected in their BMI values ($p < 0.05$). Mean activity counts were lowest when children were watching cartoons and highest when running in a sprinting game (Fig 1 and Table 2).

### Comparing MW8 with GT3X revealed significant correlations in activity counts at their respective positions, despite different operating scales

The GT3X's "vector magnitude" producing proportionally greater values than those of the uniaxis MW8 (Table 2). When data from the same brand and position were pooled across all activities, significant correlations were detected between counts measured with the MW8 and GT3X for wrist-worn (rho = 0.94) and hip-worn devices (rho = 0.90) (all $p < 0.001$, Fig 3). Averaging across all six behaviours, the mean counts/30s were significantly greater for the wrist position compared to the hip-position within the respective brands (MW8: wrist: 438.6 ± 537 counts versus hip: 300.8 ± 464.3 counts; GT3X: wrist: 3165.3 ± 3357.6 counts versus hip: 959.3 ± 937.8 counts, all $p < 0.001$). The mean activity counts/30s at wrist- and hip positions simultaneously measured with GT3X and MW8 for each behavioural activity are reported (Table 2) and visualized as boxplots (Fig 4). The boxplots reveal that the three

**Table 1. Descriptive characteristics of the participants (n = 30). Shown are means ± SD.**

|  | Males (n = 20) | Females (n = 10) | All (n = 30) |
|---|---|---|---|
| Age (years) | 3.5 ± 0.1 | 3.6 ± 0.1 | 3.5 ± 0.1 |
| Weight (kg) | 17.1 ± 1.6 | 15.3 ± 1.0* | 16.5 ± 1.7 |
| Height (cm) | 102.8 ± 3.1 | 98.0 ± 3.6 | 101.3 ± 4.0 |
| BMI (kg/m²) | 17.2 ± 1.5 | 15.3 ± 1.1* | 16.5 ± 1.7 |
| Waist circumference (cm) | 53.2 ± 2.8 | 48.5 ± 12.9 | 51.7 ± 7.9 |

*Significantly different between females and males; $p < 0.05$.

**Table 2. Mean ± SD of activity counts (counts/30 seconds) for each activity ranked from highest to lowest intensity measured at wrist-level and hip-level with ActiGraph GT3X and MotionWatch 8. Mean ± SD of the duration spent in activity is consistently similar.**

|  | N | Duration (min) | Wrist-worn | | Hip-worn | |
|---|---|---|---|---|---|---|
|  |  |  | ActiGraph GT3X (counts/30 s) | MotionWatch 8 (counts/30 s) | ActiGraph GT3X (counts/30 s) | MotionWatch 8 (counts/30 s) |
| Vigorous activity | 30 | 9.7 | 9017.8 ± 2745.2 | 1419.2 ± 457.5 | 2240.0 ± 586.6 | 1172.3 ± 357.9 |
| Moderate activity | 30 | 9.9 | 4677.3 ± 1875.3 | 672.2 ± 309.8 | 1964.7 ± 753.4 | 533.6 ± 250.5 |
| Light activity | 30 | 10.0 | 2359.8 ± 818.6 | 291.1 ± 143.9 | 1044.1 ± 550.4 | 80.3 ± 85.6 |
| Sedentary crafts | 30 | 10.0 | 1329.0 ± 736.1 | 128.9 ± 95 | 291.8 ± 317.7 | 16.0 ± 29.3 |
| Recumbent listening | 30 | 9.9 | 998.0 ± 1198.2 | 110.7 ± 142.1 | 253.5 ± 479.1 | 19.9 ± 47.4 |
| Sedentary screen time | 30 | 9.9 | 540.6 ± 724.8 | 62.9 ± 97.4 | 137.2 ± 260.3 | 10.2 ± 30.9 |

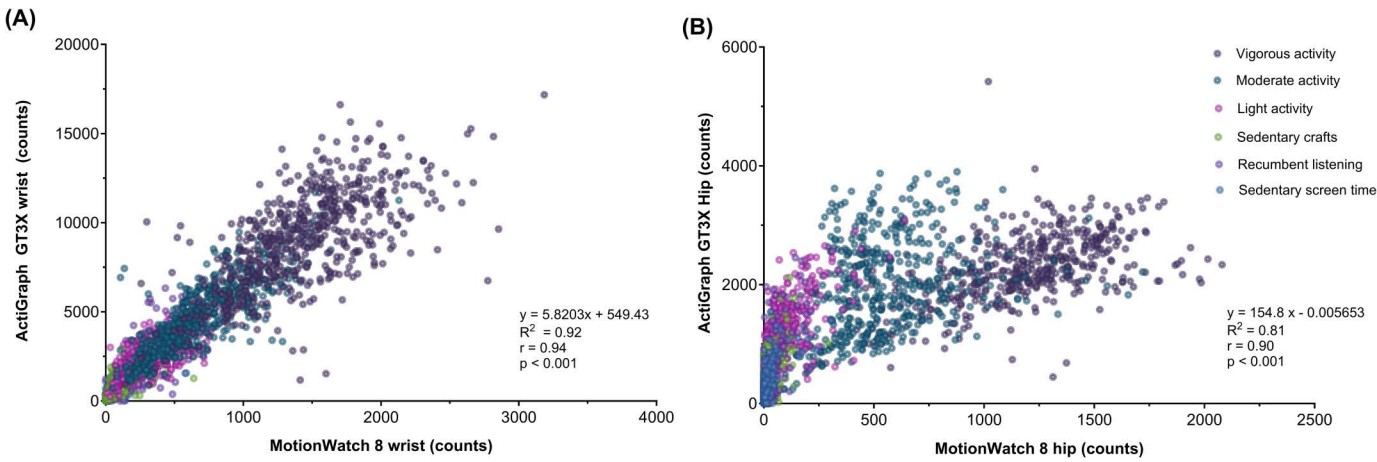

**Fig 3. Relationship between activity counts/30s in 3-year-old children (n = 30) during six activities ranging from 'Motionless alert' to 'Vigorous PA' for MotionWatch 8 (MW8) and ActiGraph (GT3X) at A) wrist position and B) hip position.** The six activities are colour-coded in the graphs.

physically mobile behaviours (light, moderate, vigorous activities) had a narrow within-class variation, enabling to separate each from one another. In contrast, the three physically stationary behaviours (recumbent listening, sedentary crafts, sedentary screen time) showed large interquartile ranges and overlap between classes, regardless of position or brand.

### Collapsing behavioural classes indicated outstanding accuracy thresholds for 'mobile PA' and excellent accuracy thresholds for 'stationary PA' intensities

OvR ROC-AUC analysis were carried out four times per device (VPA, MVPA, LMVPA and MOA). Sensitivity ranged from 89–95% and specificity from 90–94% for MVPA and VPA of wrist-worn devices. Sensitivity for *mobile* PA composite against *stationary* PA was 88% and 91%, and specificity 86% and 83% for wrist-worn MW8 and GT3X, respectively. AUC ranged from 0.95–0.98. Sensitivity for MOA (watching cartoon) against the rest was 75% and 77%, and specificity 82% and 85% for wrist-worn MW8 and GT3X, respectively. AUC was 0.86 and 0.87, respectively (Table 3).

Sensitivity ranged from 91–96%, and specificity from 81–98% for MVPA and VPA of hip-worn devices. Sensitivity for *mobile* PA composite against *stationary* PA was 85% and 91%, and specificity 91% and 89% for hip-worn MW8 and GT3X, respectively. AUC ranged from 0.90–0.98. Sensitivity for MOA (watching cartoon) against the rest was 64 and 75%, and specificity 87 and 78% for hip-worn MW8 and GT3X, respectively. AUC was 0.81 and 0.86, respectively (Table 4).

### Discussion

The motivation of conducting this calibration study came from intending to use the daily actigraphic data not only for sleep patterns but also for PA monitoring in children of an ongoing prospective birth cohort. This study aimed at providing calibration data for 3-year-old children performing various play activities and passive behaviours by means of count cut-off thresholds from wrist-worn MW8 and GT3X activity counts/30s. Additional purposes included comparison of classification accuracy between concomitantly worn wrist- and hip-mounted devices using sensitivity, specificity and ROC-AUC. Our results show that our

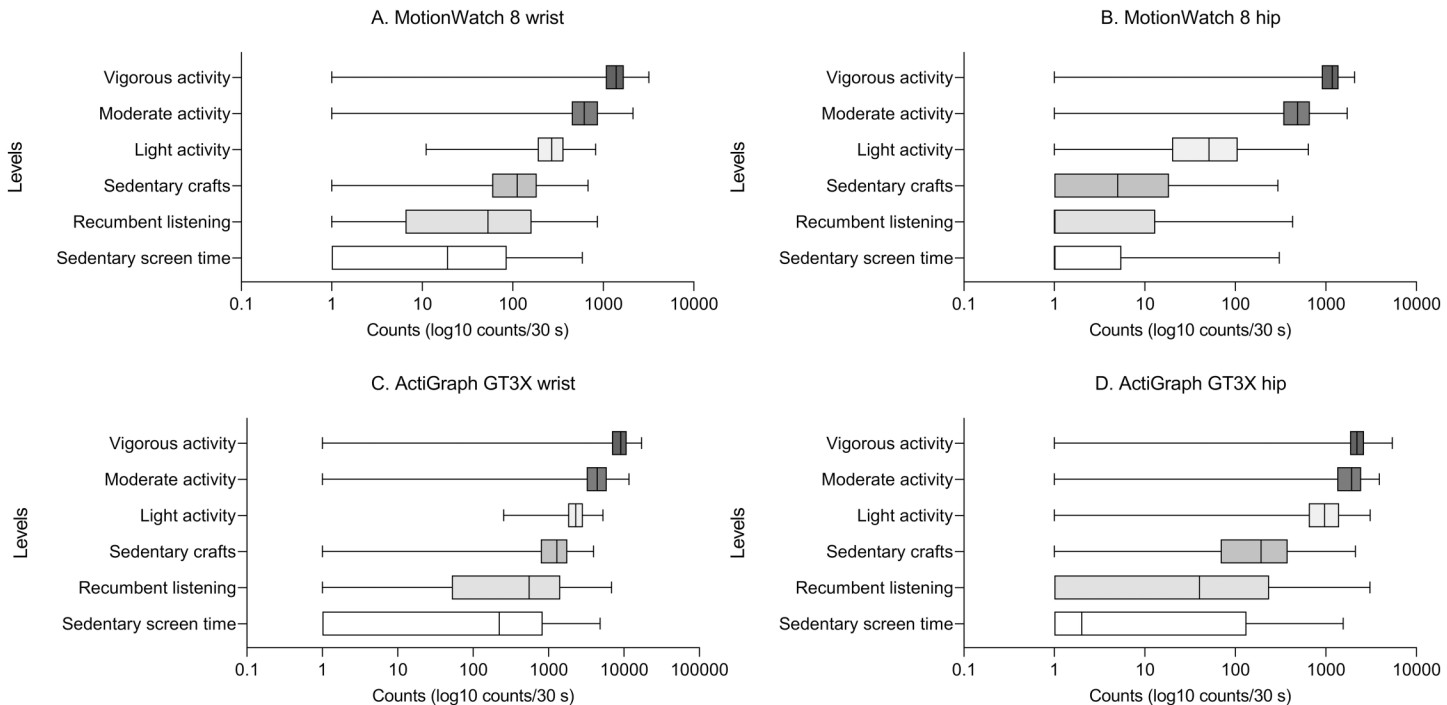

**Fig 4. Boxplots for six different intensity levels measured in 3-year old children with A) wrist-worn MotionWatch 8; B) hip-worn MotionWatch 8; C) wrist-worn ActiGraph GT3X; and D) hip-worn ActiGraph GT3X.** A value of '1' was added to every count in order to increase the visibility of the lower end of the scale, in particular for the stationary, calmer activities through a logarithmic scale. Zero movement over 30s epoch is not uncommon, for example during a break. Box contains 50% of values (Interquartile range [IQR] with values between Q1 and Q3 (25th–75th percentile), mid-line represents median, Whiskers represent min-max ([Q1 − 1.5 * IQR] − Q3 + 1.5 * IQR]).

**Table 3. ROC-AUC analysis for wrist-worn MotionWatch 8 and ActiGraph GT3X devices after collapsing categories. See details for collapsed categories in Fig 2.**

| | | Wrist- worn MotionWatch 8 (MW8) | | | | Wrist-worn ActiGraph (GT3X) | | | |
|---|---|---|---|---|---|---|---|---|---|
| | Intensity class | Cut-off value (counts) | Sensitivity (%) | Specificity (%) | AUC (95% CI) | Cut-off value (counts) | Sensitivity (%) | Specificity (%) | AUC (95% CI) |
| **Mobile PA** | VPA[1] | >787 | 93 | 94 | 0.98 | >4607 | 95 | 90 | 0.97 |
| | | | | | (0.97 to 0.98)* | | | | (0.97 to 0.98)* |
| | MVPA[2] | >408 | 90 | 93 | 0.97 | >3038 | 89 | 93 | 0.97 |
| | | | | | (0.97 to 0.98)* | | | | (0.96 to 0.98)* |
| | LMVPA[3,#] | >215 | 88 | 86 | 0.95 | >1782 | 91 | 83 | 0.95 |
| | | | | | (0.94 to 0.96)* | | | | (0.94 to 0.95)* |
| **Stationary PA** | SED[4] | 118–215 | NA | NA | NA | 1148–1782 | NA | NA | NA |
| | MOA[5] | <118 | 75 | 82 | 0.86 | <1148 | 77 | 85 | 0.87 |
| | | | | | (0.84–0.87)* | | | | (0.86–0.89)* |

*p < 0.001.

[#]LMVPA: Separates mobile physical activities (Mobile PA) from stationary activities (Stationary PA).

[1]VPA: Vigorous PA versus all other activity levels.

[2]MVPA: Moderate-vigorous PA versus all other activity levels (light, sedentary [handcraft + recumbent], motionless).

[3]LMVPA: Light moderate vigorous PA versus the rest (sedentary [handcraft + recumbent], motionless).

[4]SED: Sedentary range: above MOA threshold and below LMVPA threshold.

[5]MOA: Motionless-alert [Screentime] versus all other activity level.

**Table 4.  ROC-AUC analysis for hip-worn MotionWatch 8 and ActiGraph GT3X devices after collapsing categories. See details for collapsed categories in Fig. 2.**

| | Intensity class | Hip-worn MotionWatch 8 (MW8) | | | | Hip-worn ActiGraph (GT3X) | | | |
|---|---|---|---|---|---|---|---|---|---|
| | | Cut-off value (counts) | Sensitivity (%) | Specificity (%) | AUC (95% CI) | Cut-off value (counts) | Sensitivity (%) | Specificity (%) | AUC (95% CI) |
| Mobile PA | VPA[1] | >637 | 92 | 94 | 0.98 | >1509 | 91 | 81 | 0.90 |
| | | | | | (0.97 to 0.99)[*] | | | | (0.89 to 0.91)[*] |
| | MVPA[2] | >214 | 96 | 98 | 0.99 | >1006 | 94 | 85 | 0.96 |
| | | | | | (0.99 to 0.99)[*] | | | | (0.95 to 0.96)[*] |
| | LMVPA[3,#] | >46 | 85 | 91 | 0.95 | >631 | 91 | 89 | 0.96 |
| | | | | | (0.94 to 0.95)[*] | | | | (0.95 to 0.96)[*] |
| Stationary PA | SED[4] | 22–46 | NA | NA | NA | 183–631 | NA | NA | NA |
| | MOA[5] | <22 | 64 | 87 | 0.81 | <183 | 75 | 78 | 0.86 |
| | | | | | (0.79–0.83)[*] | | | | (0.84–0.87)[*] |

[*]p < 0.001.

[#]LMVPA: Separates mobile physical activities (Mobile PA) from stationary activities (Stationary PA).

[1]VPA: Vigorous PA versus all other activity levels.

[2]MVPA: Moderate-vigorous PA versus all other activity levels (light, sedentary [handcraft + recumbent], motionless).

[3]LMVPA: Light moderate vigorous PA versus the rest (sedentary [handcraft + recumbent], motionless).

[4]SED: Sedentary range: above MOA threshold and below LMVPA threshold.

[5]MOA: Motionless-alert [Screentime] versus all other activity levels.

cut-off points corroborate the separation of physically *mobile* behaviours from physically *stationary* behaviours (specificities of 83%-86% for wrist-worn cut-offs), which is in line with results of earlier studies, despite using different measures and criteria of classifying PA intensities [10,12,32,49]. The movement intensities differed not just by behaviour but also by type of device and body position. The count cut-offs from both devices (MW8, GT3X) at both positions (wrist, hip) showed outstanding accuracy (range: 0.95–0.98) for all *mobile* PA classes. Nevertheless, the classification accuracy for 'motionless alert' against all other classes showed a fair accuracy (range: 0.81–0.87).

## Observational method

Direct observation was the choice of criterion to calibrate activity counts because it avoids interpretation errors associated with MET conversions or errors associated with extrapolation from atypical activities (walk/dance to a certain pace) to free-living behaviours [60]. The pilot phase has been valuable to adjust activities to the children's age, i.e. children refused to dance to the beat of music. Instead this moderate activity was changed into a moderately paced walk with their parent or playing with miniature floorball. The use of pre-determined behaviours allowed us to classify sets of activities, i.e. as stationary. No previous studies have developed activity intensity thresholds for wrist- and hip-worn GT3X models at 30s-epochs in vector mode (VM). But thresholds have been published for GT3X models at 5s-epochs using a sampling rate of 30Hz and wrist/hip positions [12] and for the Actiwatch (Minimitter) at 1-min epochs and a 32 Hz sampling rate [58]. While the present study employed single child observations over six directed activities, Johansson et al. (2016) [12] and Finn and Specker (2000) [58] employed video recording during free activities of groups of children and subsequently assigned behaviours according to the 'Children's Activity Rating Scale' [61,62]. The activities in the present study included similar types as in the other studies, i.e. watching cartoons and drawing; refrained from other types, i.e. dancing, outdoor: and included different types, i.e. sprinting game, floor-ball. [12]. Compared to Johansson's et al. (2016), the intensity

thresholds established for GT3X in VM in the present study were higher. This was expected from an epoch five times as long, confirming thresholds to be epoch-specific in addition to being age-specific [12].

## Types and terminology of movement behaviours

We found substantial overlap in activity counts among the three physically *stationary'* activities as evident from the sizable variation in the interquartile range of the box plots. Already two decades ago, correlations between activity counts and oxygen consumption were reported for rest/structured activities and free play (r = 0.82 and r = 0.66, respectively) in 3–5-year old children [13], whereupon stationary behaviours, also termed 'physical *inactivity'* became recognised as a distinct construct [60]. The Children's PA Research Group at the University of South Carolina has made significant contributions to better understand PA behaviours in children [21,63]. Among those, most notably the development of observational instruments to assess PA context-specific, and the implementation of accelerometry cut-off points in settings of different age groups. The overlap of activity counts for stationary behaviours implies that non-stationary PA can relatively clearly defined in future large-scale data-driven approaches. Cut-off points can be applied in longitudinal actigraphic data collected in sleep research and PA analysed while sleep is treated as a different vigilance state than quiet awake. In the future, PA and sleep analysis of actigraphic/accelerometer population data should be combined as movement-based 24-h behaviour. The correlation between PA activity and oxygen consumption [13] justifies to address circumstances and conditions surrounding low PA activities, for example the transition from home to school or living in city, urban or rural areas. Activity monitoring can be helpful for identifying children in need of improving their fitness and tailoring interventions for both, sleep and PA with staff and family support.

The current study adopted the term 'motionless alert' instead of using 'rest/resting' because the term 'rest' is inconsistently used [64]. In chronobiology, 'rest' covers 'sleep', including sleep-defining stages 'time-in-bed' 'latency to sleep', and 'sleep period' (motionless body with an active brain). The Sedentary Behaviour Research Network has published consensus definitions to standardise terminology for sedentary behaviours using 'posture' information and 'energy expenditure' [65]. They also adopted the term 'stationary' and suggested to apply 'behaviour' when the context is known, and 'time' in the absence of context, which is a sensible linguistic differentiation. We advocate to use terminology that is applicable not only for PA but for different disciplines such as chronobiology, sleep and epidemiology.

An important objective of observing the behaviour while measuring movement intensities was to test the discrimination power for 'motionless alert' behaviour, such as watching a cartoon, from all other classes. Although acceptable, 'motionless alert' from the wrist-worn MW8 and GT3X cut-offs showed only fair sensitivity (75% and 77%), while sensitivity was even lower for hip-worn cut-off points. The better outcome of the *mobile* PA classes over *stationary* PA classes can be explained by greater density of counts with much narrower ranges due to more steady, continuous activity during 'sprinting, 'floor ball' and 'playing on the floor'. In comparison, the *stationary* PA classes, and motionless alert in particular, included children engaged in mental tasks (listening, concentrating, observing) with the occasional, discontinued burst of movements while sitting still, for example when pointing or changing of position. Whether distinguishing the different stationary behaviours for time spent in low PA per day is useful can be questioned, but distinguishing motionless alert from sleep is necessary. Visual inspection, time-stamp markers or diary entries (night sleep, nap) have been proven to work in chronobiology and sleep science.

## Sensory modes and body positions

In direct comparison, the two device models systematically scaled count quantities differently irrespective of placement (wrist/hip). Since all manufacturers calibrate their models at factory for optimal recording performance suited for certain body position, there is an expected imbalance in counts related to body positions [40,66]. For example, the MW8 offers different recording options such as uni- or triaxial mode, specific for wrist movements or body movements, respectively. Its uni-axial mode has been calibrated for wrist position, making use of omni-rotary arm movements [67], and sensitivity thresholds were determined and validated for sleep analyses algorithms (S1 Table). Its tri-axial mode produces VM/epochs that is best suited for the trunk, i.e., the hip, but not for measuring sleep nor light [54]. We mounted a second MW8 in uni-axial mode at the hip position to see how the same mode scales wrist against hip position, accepting its compromised status of reduced resolution. Lower activity counts were found for all behavioural activities at hip position compared to the wrist not only for MW8 but also GT3X. Differences in intensity output using two raw (with bandpass filters) acceleration metrics (Mean amplitude deviation, Movement acceleration intensity) also showed higher values for the wrist position compared to hip position when measured simultaneously in children 8–13 years old [68]. The GT3X model, which was set to tri-axial mode as recommended, generated approximately 6 to 10 times higher values in VM mode than the uni-axial MW8 mode, regardless of position. The challenge with recording and processing raw 3-dimensonal *longitudinal* data is their volume - in the TeraByte range - and the need to develop approaches on how to visualise 3-dimensional patterns, most summarise them into a single vector/epoch [69]. Taken together, users need to be aware of these facts and decide accordingly on sensor modes, pre-filter/raw settings and algorithms when implementing activity monitoring into longitudinal recordings.

## Within model comparison of cut-off points

The MW8 has previously been validated to measure PA in older adults (S1 Table) [47] and in 9–13 year old children [49]. Lin et al.'s (45) protocol differed from our study in that they captured data from the dominant wrist position of older children. We preferred the non-dominant wrist position because it has been shown to produce less misclassification for sedentary behaviours [42]. Despite the differences, our results are in line for *mobile* PA activities with Lin et al. [49], who reported a good ability of the device to differentiate light from moderate-to-vigorous activity (MVPA >371.5 counts/30 s), and moderate from vigorous activity (VPA >859.5 counts/30 s). Our boundaries for VPA is a little lower (MVPA > 408 counts/30s; VPA >787 counts/30s), likely as a result of age-related differences in motor development and muscle strength. In Johansson et al. [12], authors argued for age-specific thresholds since they found that 4-year-old preschool children, who wore a GT3X triaxial on their non-dominant wrist, produced generally higher intensity thresholds per 5s-epoch in comparison to toddlers of 2 years [19]. Differentiation of sedentary from light activity was much greater in the study by Lin et al. (45), given the much lower sedentary threshold score (SED < 32 counts/30s) compared to the present study (SED < 215 counts/30s, MOA < 118 counts/30s). This is likely related to Lin et al. (45) prioritizing higher specificity at the cost of lower sensitivity in order to minimise false positives due to arm movements. Johansson et al. [12] used the opposite adjustment - towards higher sensitivity at the expense of a lower specificity for the SED threshold in order to avoid underestimating the time spent in SED. In the present study we used a step-approach with the upper boundary for SED being the lower boundary of *mobile* PA classes against all combined *stationary* PA classes, which includes arm movements, i.e. when sitting and doing crafts.

All threshholds were reported as the greatest sum of the sensitivity and specificity, without prioritising specificity nor sensitivity.

## Strength and limitations

One of the strengths is the comparative study design that included two research-grade devices (MW8, GT3X) worn in parallel at two body positions (wrist and hip) in a young age group (3 years). This allowed us to point out distributional regularities, similarities and differences for the devices and positions, knowledge that is important to consider when planning a study or when comparing data collected with these devices. In addition, the use of a 30 second epoch matches the calibration of MW8 devices for sleep assessments and meets the requirements for longitudinal, long-term actigraphic data collection for circadian rhythm/sleep assessments. The present calibration allows PA analysis from measurements of physical activity in combination with circadian/sleep assessments. The purpose of it is a multi-modal outcome with relevant information for trajectories of 24-h sleep and PA sufficiency to be comparable and interpretable with other studies and contribute to charts of 24h sleep and PA percentile curves. Half a minute may seem very long for sport activities but human behaviours under real situations last for several minutes, in particular conversations, meal and play times, sedentary activities like handcrafts, drawing or screen time. On the other hand, vigorous activities over the sequence of 10 minutes were not uninterrupted. Instead, there were periods of short light level movements.

Another strengths is the application of a commonly used classification method on directly observed behaviours to quantify intensity levels from which count cut-off points were determined. The intensity thresholds were chosen based on the best compromise between sensitivity and specificity, since prioritising sensitivity for 'physically stationary' may inflate sedentary time, when it is actually time spent in light mobile PA. Prioritsing specificity for stationary PA may overestimate 'physically mobile' and underestimate time spent sedentary. Our approach is a compromise as are those of others [5]. Our calibration may be applicable to existing population datasets since it allows to derive scaling coefficients. When other studies used 5 second epochs these can be re-scaled to match the 30s epoch for comparison, but this conversion needs verification as deliniated by Orme et al. [70]. The dataset generated during the current study is freely available under figshare.

The absence of indirect calorimetry [71] could be seen as a limitation, but oxygen consumption has been measured indirectly in 4-year old preschool children and activity intensity levels at free play correlated well with oxygen consumption, yet it has been discussed to have limitations due to the delayed rate of change with a change in activity levels [13]. We are aware that our thresholds have not undergone cross-validation and all of the observed activities were performed indoors, although resembling everyday habits of a three-year-old child. Outdoor activities were deliberately left out because of the high seasonal variability in outdoor activities in Northern Sweden. Further work is required to examine season-based behavioural outdoor activity patterns.

## Conclusions

In conclusion, MotionWatch 8 and the ActiGraph GT3X underwent calibration processing simultaneously on the non-dominant wrist and hip based on direct observation of six naturally-occuring behavioural activities in healthy, 3-year-old children. ROC-AUC curves revealed cut-off points with outstanding (Sprinting, Floorball/Walk and Playing on the floor) and excellent (Sedentary screen time) discrimination power. The accuracy indicates that the calibrated cut-off points can be adequately used to determine time spent in different activity levels. When applying the cut-off points, five calibration factors need to be matched (i) age(ii)

the device-specific sensor mode(iii) epoch (scaling coefficient, if different)(iv) body position, and (v) behaviours equivalent to occurring in natural settings. The dataset is freely available.. It is hoped to be useful for the community interested in a 24-hour movement-related approach, integrating PA intensities into projects on sleep, circadian rhythms and light exposure.

## Supporting information

**S1 Table. Highlighting some of the different approaches taken from sports medicine and chronobiology/sleep, with references of published examples (References Supplementary to** S1 Table**: 42 articles).**
(DOCX)

**S2 Table. Receiver operating characteristics curve (ROC) analysis in One-vs-One scheme, comparing pairwise combination of adjacent behavioural classes as described in** Fig 2a**.** Cut-off values, sensitivity, specificity and accuracy (AUC) are reported from 'vigorous' to 'motionless alert'. 'Sedentary crafts' and 'Recumbent listening' were merged into 'Sedentary (active)'. 'Sedentary screen time is categorised as 'Motionless alert'.
(DOCX)

**S3 Fig. Duration in minutes of cumulated wear-time of devices (MW8, GT3X) simultaneously worn at wrist and hip positions while performing each of the six behaviours.** The epoch for storing the activity counts was 0.5 min.
(DOCX)

**S4 Fig. ROC curves for the wrist-worn Motionwatch 8.** OvR ROC curves illustrating the results in Tables 3 and 4 in the main text for: **A**) vigorous physical activity (VPA), **B**) moderate-vigorous physical activity (MVPA), **C**) light moderate vigorous physical activity (LMVPA) and **D**) motionless-alert (MOA).
(DOCX)

**S5 Fig. ROC curves for wrist-worn GT3X.** OvR ROC curves illustrating the results in Tables 3 and 4 in the main text for: **A**) vigorous physical activity (VPA), **B**) moderate-vigorous physical activity (MVPA), **C**) light moderate vigorous physical activity (LMVPA) and **D**) motionless-alert (MOA).
(DOCX)

**S6 Fig. ROC curves for hip-worn Motionwatch 8.** OvR ROC curves illustrating the results in Tables 3 and 4 in the main text for: **A**) vigorous physical activity (VPA), **B**) moderate-vigorous physical activity (MVPA), **C**) light moderate vigorous physical activity (LMVPA) and **D**) motionless-alert (MOA).
(DOCX)

**S7 Fig. ROC curves for hip-worn GT3X.** OvR ROC curves illustrating the results in Tables 3 and 4 in the main text for: **A**) vigorous physical activity (VPA), **B**) moderate-vigorous physical activity (MVPA), **C**) light moderate vigorous physical activity (LMVPA) and **D**) motionless-alert (MOA).
(DOCX)

## Acknowledgments

The authors thank all children and their parents for their participation. Also, we thank Anna Tellström and Rebecca Rönnholm for their great assistance in the data collection; Tobias Stenlund for all organisational help with the e-health laboratory; Patrik Wennberg for lending

us the two Actigraph GT3X devices and the accompanying temporary license; and Richard Lundberg for his administrative support with the NorthPop database.

## Author contributions

**Conceptualization:** Jonas Sandlund, Magnus Domellöf, Katharina Wulff.

**Data curation:** Magnus Domellöf, Katharina Wulff.

**Formal analysis:** Daniel Jansson, Rikard Westlander, Magnus Domellöf, Katharina Wulff.

**Funding acquisition:** Jonas Sandlund, Christina E. West, Magnus Domellöf, Katharina Wulff.

**Investigation:** Daniel Jansson, Rikard Westlander.

**Methodology:** Jonas Sandlund, Magnus Domellöf, Katharina Wulff.

**Resources:** Jonas Sandlund, Christina E. West, Magnus Domellöf, Katharina Wulff.

**Supervision:** Magnus Domellöf, Katharina Wulff.

**Writing – original draft:** Daniel Jansson, Rikard Westlander, Katharina Wulff.

**Writing – review & editing:** Daniel Jansson, Jonas Sandlund, Christina E. West, Magnus Domellöf, Katharina Wulff.

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
