## [Decision Letter · Decision Letter 0]

9 Jun 2024

PONE-D-24-07355Behaviour-based movement cut-off points in 3-year old children comparing wrist- with hip-worn actigraphs MW8 and GT3XPLOS ONE

Dear Dr. Wulff,

Thank you for submitting your manuscript to PLOS ONE. After careful consideration, we feel that it has merit but does not fully meet PLOS ONE’s publication criteria as it currently stands. Therefore, we invite you to submit a revised version of the manuscript that addresses the points raised during the review process.

We look forward to receiving your revised manuscript.

Kind regards,

Duncan S Buchan

Academic Editor

PLOS ONE

Reviewers' comments:

Reviewer's Responses to Questions

**Comments to the Author**

1. Is the manuscript technically sound, and do the data support the conclusions?

Reviewer #1: Partly

Reviewer #2: Yes

2. Has the statistical analysis been performed appropriately and rigorously? 

Reviewer #1: Yes

Reviewer #2: Yes

3. Have the authors made all data underlying the findings in their manuscript fully available?

Reviewer #1: Yes

Reviewer #2: Yes

4. Is the manuscript presented in an intelligible fashion and written in standard English?

Reviewer #1: Yes

Reviewer #2: Yes

5. Review Comments to the Author

Reviewer #1: Dear Editor, Dear Author,

The article "Behaviour-based movement cut-off points in 3-year old children comparing 4 wrist- with hip-worn actigraphs MW8 and GT3X" is a study focusing on calibration of activity counts of motor behavior measures simultaneously with two devices.

Thank you for the opportunity to review this manuscript. The article deals with a relevant and contemporary issue and is based on a good scientific methodological quality. The research question at hand and the methodological approach are discussed comprehensively. Nevertheless, I have some concerns that should be addressed by the authors before publication.

General

- Please use PA instead of physical activity throughout the whole manuscript, since you introduce the abbreviation at the beginning.

- Please improve quality of all figures.

Abstract

1. Line 46, page 2: What is meant with rigorous calibration? Is this something different than just a calibration?

2. Line 52, page 2: Are the six activities allocated to different intensities? Please specify.

3. Line53, page 2: I wonder if the the time of each activity is too long, especially for vigorous intensities when participants are three year old? In Line 53 (page 2) you also mentioned sprinting about 10 minutes. Are 3-year old children able to do this for 10 minutes?

4. Line 54, page 2: What do you mean with directly observed? You mentioned this the term but we wonder what exactly is meant with this and couldn’t find any more information in the manuscript neither in the results section, nor in the discussion.

5. Line 61, page 2: Why classifying into mobile and stationary? Maybe you can add one sentence to clarify this already in the abstract.

6. Line 65, page 2: I wonder what you mean with context information. Please specify.

Introduction

General:

- Throughout the whole introduction, sentences are very long. I would recommend, separating long sentences (especially Line 78-84, page 3; Line 129-136, page 5).

- The introduction is very long. I would recommend shorten it. The history of accelerometers is not that important in my opinion (Line 73, page 3). Further, some information could be more summarized. Please critically check the introduction for shortening.

1. Line 83, page 3: World Health Organization (WHO)

2. Line 140, page 5: What is the intention to assess context-related movements and distinct between mobile and stationary? From a public health perspective, it is necessary and more common to focus on intensities, as especially MVPA leads to health benefits and LPA does not impact health status in that way.

Method

1. Page 6, Line 150: Healthy 3-year-old children. What does that mean? Without disabilities? Without illness? I would recommend rephrasing the sentence in a Subject-first language (3-year old children without…).

2. Page 7, Line 170: Did you use expected MET-values? How did the classification of the activities to the different intensities took place?

3. Page 7, Line 181: On which basis did you chose the activities?

4. Page 8, Line 197: I wonder why you didn’t validate your data with the heart rate when you have the data anyway? Did you do something to validate the data? Or just the correlation between two similar devices?

5. Page 8, Line 211. Please remove the space between 250 g.

6. Page 9, Line 219: The raw data of the accelerometer are accelerations (g as unit). How do you get counts from the acceleration?

7. Page 9, Line 225: proprietary software – what is this? I am not sure if everybody know what this means. Please contextualize.

8. Page 9, Line 226: what is the behavioral observation protocol. Please describe this in more detail. This term is frequently in your manuscript but for me it is not sure what is meant with that.

9. Page 9, Line 231: Why didn’t you use test-retest to assess the reliability? How did you assess the reliability? Known studies often focus on validity and reliability in the context of calibration.

10. Page 10, Line 241: observer-based behaviors – what is this? Please contextualize.

11. Page 10, Line 246: I wonder why you make a distinction between mobility and stationary behavior when the common classification are the intensity levels.

12. Page 10, Line 252: on which basis did you do that? MET-values?

Results

1. Do you have the individual raw data/activity counts of each conducted activity?

2. Page 16, Table 3: Cut-off values per which second?

Discussion

1. General: Some paragraphs are difficult to understand. Maybe it would be helpful to add some subheadings to help the reader to go through the discussion (e.g., cut-off values, sensor position, …) see also (Beck et al., 2023).

2. Page 16, Line 382: Again, what is the observation technique? Are there no results concerning this?

3. Page 16, Line 383: Children’s activity rating scale  in your study? The sentence is not clear to me.

4. Page 19, 390 ff: Is this also a phenomenon found in actual literature as you mention “Already two decades ago”?

5. Page 20, Line 397: You did not mention the qualitative-quantitative combination before. Please state this also in the methods section/abstract.

6. Page 20, Line 411- 422: In this section, you just describe your results. Please shorten this and state the main finding and discuss this.

7. Page 21, Line 430: Why didn’t you mention your motivation for this study at the beginning of the discussion?

8. Page 21, Line 430 ff: I suggest restructure this section. What was your main result? Further there are many statements not supported by literature (e.g. Since all manufacturers calibrate their models at factory for optimal recording performance suited for certain body position, there is an expected imbalance in counts related to body positions). Which implications does this have on your results?

9. Page 22, Line 443: Compare this with existing studies (e.g. Beck et al., 2023)

10. Page 22, Line 452 ff: I suggest restructure this section and firstly mention your results, then discussing. Further, in this section wearing positions (method) are discussed. It

11. Page 23, Line 475: Please add a subheading  Strengths and Limitations

12. Page 23, Line 475: I would prefer restructuring the strength-section. Actually, you are not contextualizing your first strength - “One of the strengths is the study design that included two research-grade devices (MW8, GT3X) worn in parallel at two body positions (wrist and hip) in a young age group (3 years)” – why is this a strength of your study? Maybe you could pronounce this in one more sentence.

Reference:

Beck, F., Marzi, I., Eisenreich, A., Seemüller, S., Tristram, C., & Reimers, A. K. (2023). Determination of cut-off points for the Move4 accelerometer in children aged 8–13 years. BMC Sports Science, Medicine and Rehabilitation, 15(1). https://doi.org/10.1186/s13102-023-00775-4

Reviewer #2: A nicely conducted study. A few recommendations to improve its readability:

INTRODUCTION

• Line 72-73: Repeated use of "segregated" and "integrated" could be streamlined for clarity. SUGGESTED REVISION: "Movement-related assessments of habitual activities, such as physical activity (PA) levels or timing of sleep/circadian rhythms, have typically been segregated across disciplines like epidemiology, sports medicine, rehabilitation, and chronobiology [1–3]."

• Line 72-75: SUGGESTED REVISION: "Assessments of habitual activities like physical activity (PA) levels or sleep/circadian rhythms have often taken a segregated approach in fields such as epidemiology, sports medicine, rehabilitation, and chronobiology [1–3]. Historically, terminology also developed independently, with 'actimeter' or 'actigraph' used in sleep/chronobiology and 'accelerometer' in sport/physical activity (Tab S1)."

• Line 75: SUGGESTED REVISION: "Similarly, evidence regarding the combination of movement behaviours over a 24-hour period using compositional analyses [4] is uncommon but growing [5], with emerging studies [6–8]."

• Line 131: "https://www.katlab.org/ [under people]". SUGGESTED REVISION: "https://www.katlab.org/ [under 'people'], www.northpop.se"

• Context and Rationale: The introduction could benefit from a more explicit rationale for why comparing wrist- and hip-worn actigraphs is essential. While the text touches on different devices and algorithms, it should clarify the practical implications of these comparisons for assessing PA and sleep in children. i.e. explain why both wrist and hip placements are necessary and what specific insights are gained from comparing these placements.

• Research Gap: The introduction mentions the need for more studies but does not clearly state what specific gap this study aims to fill beyond general calibration. Clarify what unique aspect of children's activity measurement this study addresses.

• Detail on Current Standards and Practices: While the text mentions various guidelines and recommendations, it could provide a clearer connection between these guidelines and the specific challenges or limitations this study aims to overcome. For example, how current guidelines fail to account for the differences between wrist- and hip-worn devices in practical terms.

METHODS

• Participant Recruitment and Criteria: Lines 148-152: The recruitment criteria are clear, but additional context on the rationale for these criteria would be beneficial. Why specifically exclude children with chronic diseases or those outside the normative weight range?

• Lines 158-159: A brief mention of how the children's consent was obtained, beyond the legal guardians' consent, would be useful.

• Lines 166-167: A pilot is a crucial step for validating the methods, but the results of this pilot study are not discussed. A brief mention of any adjustments or findings from the pilot could enhance the credibility of the methods.

• Line 168-172: The description of the six behaviors is clear, but the introduction of "vigorous," "moderate," and "light" activity could be linked more explicitly to how these terms are operationalized in the study. Consider briefly defining these terms in this section for clarity.

• Lines 220-221: It would be helpful to reference specific studies or data supporting the Epoch Length choice.

STATISTICS

• Line 233: "… using a standard Students Unpaired T-test.". The choice of the Student's Unpaired T-test is standard for comparing means between two independent groups. However, if the sample size is small, a mention of checking assumptions of normality and equal variances (e.g., via Levene's test) would strengthen the methodological rigor.

• Line 235: "The two-tailed Pearson product-moment correlation was used ...". Pearson's correlation assumes linear relationships and normally distributed variables. Clarifying if these assumptions were tested and met would be beneficial. If not, a non-parametric correlation test (e.g., Spearman's rank correlation) might be more appropriate.

• Regression Models Lines 236-237: This statement lacks clarity on why different regression models are used for wrist vs. hip data. Justifying this choice with underlying data characteristics or preliminary analysis results would provide better context. Moreover, specifying the type of nonlinear regression model used (e.g., polynomial, exponential) would enhance transparency.

• Line 238: Using boxplots on a log scale is a good approach for skewed data. However, explaining why a log transformation is necessary (e.g., due to the skewed distribution of activity counts) would add clarity.

• Lines 240-250: The approach of using ROC curves to determine cut-off points is well-established. However, the explanation could be improved by: (1) Clarifying why a binarized approach is used and its advantages over multi-class ROC analysis. (2)Providing more details on how the specific behavioural categories were chosen and merged for the ROC analysis.

• Line 273: A brief rationale for choosing the Youden Index over other potential indices (e.g., F1 score, balanced accuracy) would be beneficial.

• Lines 277-282: The interpretation of ROC-AUC values is clearly stated, but ensuring that these values are consistently used throughout the analysis section will strengthen the overall analysis. Any variations or deviations from these interpretations should be noted and justified.

• There is no mention of a power analysis or justification of the sample size. Given that only 30 children were included, a brief explanation of how this sample size was determined to be adequate for the statistical tests used would be important.

• There is no discussion on how missing data were handled. Addressing whether there were any missing data points, and if so, how they were managed (e.g., imputation methods) would enhance the transparency and reliability of the results.

RESULTS

• The results section is thorough and well-structured.

DISCUSSION

• Aim and Context: (1) The aim of the study is clearly stated in the opening sentences (Lines 375-379). However, the text could be more concise, directly stating the key objectives without repetition. (2) Including a brief summary of the main findings at the beginning would help set the stage for the detailed discussion that follows.

• Comparison with Previous Studies: (1) The comparison with previous studies (Lines 379-387) is thorough but somewhat scattered. It would be beneficial to organize this section by specific themes or metrics (e.g., epoch length, device differences) to improve readability. (2) The explanation of why the thresholds in this study are higher due to the longer epoch length is informative but could be simplified for better clarity.

• Activity Counts and Behavior Distinction: (1) The discussion of overlap in activity counts among stationary activities (Lines 388-389) is clear, but it would be helpful to provide more context on the implications of this overlap for practical applications or further research. (2) The reference to historical data on correlations between activity counts and oxygen consumption (Lines 390-393) is useful, but the relevance to the current study should be made more explicit.

• Terminology and Definitions: (1) The section on terminology (Lines 398-407) is detailed but could be streamlined. The distinction between 'motionless alert' and 'rest' is important, but the explanation could be more concise. (2) The discussion on standardized definitions by the Sedentary Behaviour Research Network is relevant, but it would benefit from a clearer link to how these definitions were applied or interpreted in this study.

• Statistical Analysis and Accuracy- ROC-AUC and Sensitivity/Specificity: The explanation of why mobile PA classes showed better accuracy than stationary PA classes (Lines 423-428) is clear, but further elaboration on potential strategies to improve stationary PA classification could be beneficial.

• Device Comparison: (1) The comparison between MW8 and GT3X devices (Lines 430-451) is detailed but could be more structured. Breaking this section into sub-sections (e.g., device-specific performance, position-specific performance) would improve readability. (2) The discussion on the different calibration modes and their implications (Lines 436-450) is important but could be condensed for clarity.

CONCLUSION:

• The conclusion could be clearer by breaking down complex sentences. For example, the sentence starting with "The accuracy indicates..." (Lines 511-515) is quite dense and could be split for better readability.

• The mention of "ROC-AUC curves revealed cut-off points with outstanding, excellent and good discrimination power" (Lines 510-511) could be more specific. It would be beneficial to state which activities or behaviors correspond to each level of discrimination power.

• The conclusion could be clearer by breaking down complex sentences. For example, the sentence starting with "The accuracy indicates..." (Lines 511-515) is quite dense and could be split for better readability.

• The mention of "ROC-AUC curves revealed cut-off points with outstanding, excellent and good discrimination power" (Lines 510-511) could be more specific. It would be beneficial to state which activities or behaviors correspond to each level of discrimination power.

6. PLOS authors have the option to publish the peer review history of their article (what does this mean? ). If published, this will include your full peer review and any attached files.

**Do you want your identity to be public for this peer review?** For information about this choice, including consent withdrawal, please see our Privacy Policy .

Reviewer #1: No

Reviewer #2: No

---

## [Author Response · Author response to Decision Letter 1]

25 Nov 2024

RESPONSE TO REVIEWERS

PONE-D-24-07355

Behaviour-based movement cut-off points in 3-year old children comparing wrist- with hip-worn actigraphs MW8 and GT3X

We thank the editor and reviewers for their critical reading of the manuscript and their questions and comments to clarify phrases and procedures we described in the manuscript. Below, we have written a reply point by point, starting with +++ and changes we made in the manuscript according to the comments and suggestions are in track changes in the revised manuscript.

Reviewer #1: Dear Editor, Dear Author,

The article "Behaviour-based movement cut-off points in 3-year old children comparing 4 wrist- with hip-worn actigraphs MW8 and GT3X" is a study focusing on calibration of activity counts of motor behavior measures simultaneously with two devices.

Thank you for the opportunity to review this manuscript. The article deals with a relevant and contemporary issue and is based on a good scientific methodological quality. The research question at hand and the methodological approach are discussed comprehensively. Nevertheless, I have some concerns that should be addressed by the authors before publication.

General

- Please use PA instead of physical activity throughout the whole manuscript, since you introduce the abbreviation at the beginning.

+++ We have replaced ‘physical activity with PA throughout since abbreviation was introduced. We replaced ‘physical’ with ‘motor’ in a sentence related to sleep.

- Please improve quality of all figures.

+++ We noticed that the figures in the PDF are of poorer quality than the original figures. We anticipate that the higher quality will remain in the type-setting process.

Abstract

1. Line 46, page 2: What is meant with rigorous calibration? Is this something different than just a calibration?

+++ Rigorous is meant to underscore that we considered every part of the process to make certain that it is correct. This includes testing the sensors by putting the devices on a rotor with a constant round per second to check at the recoding epoch that the devices themselves do produce very similar counts. We made sure they arrow on the devices were always placed in the same direction on the wrist and the waist. The wrist-worn devices need to be placed either on the dominant or non-dominant wrist for comparison. None-rigorous calibration ignores these facts, which the increases systematic errors. We wanted to highlight that we are aware and have tried our best to minimise systematic errors.

2. Line 52, page 2: Are the six activities allocated to different intensities? Please specify.

+++ Yes, we have amended the abstract to include the categories.

3. Line53, page 2: I wonder if the the time of each activity is too long, especially for vigorous intensities when participants are three year old? In Line 53 (page 2) you also mentioned sprinting about 10 minutes. Are 3-year old children able to do this for 10 minutes?

+++ The length was calculated by the need to have enough data per child with an epoch of 30s (minimum 16 data points for 8 min). The vigorous activity was a sprinting game along the corridor, where they had to carry balls from a box on one side into a box at the other side. So, they had not a continuous sprint for 10 min, which would not reflect their usual behaviour. Instead they had short breaks at each side to take their breath. This is something we tested in the pilot and found that 3-year olds can do that very well without exhausting themselves too much. In terms of intensity levels, since the epoch was 30s, these short breaks would not create zero values but give a representative activity count for vigorous activity for this age. We have explained this under methods:

Here the children were asked to carry balls from a box on one side of a long corridor to another box at the other end. They had short breaks to take their breath at each side. They did not run continuously fast for 10 min.

4. Line 54, page 2: What do you mean with directly observed? You mentioned this the term but we wonder what exactly is meant with this and couldn’t find any more information in the manuscript neither in the results section, nor in the discussion.

+++ ‘Directly observed’ means we had one person, who was not involved in the execution of the activities but watching (observing) from a ‘bird’s perspective’ over each activity with a stopwatch. This person had the list of activities on a sheet of paper and kept accurate note of the start time and end time to make sure the activity lasted for 8 to10 min and can be accurately identified later in the time series of the recordings. This enabled the second person to focus entirely on the child without being distracted by checking the time. We made it more explicit under methods, see below and dedicated it a subheading in the discussion:

Under methods: …which is here termed ‘directly observed’. The ‘direct observation protocol’ entails one person acting as a ‘bystander’ watching (observing) each child’s activities from a ‘bird’s perspective’ and documenting each activity’s start and stop with a stopwatch. This enabled the second person to engage with the child without being distracted by checking the time. The documented start and stop times were essential for extracting the sequences of activities from the times series.

+++ We used this terminology in the abstract:

Time-keeping was ensured using direct observation by an observer.

The term ‘direct observation’ was used by Freedson et al (2005) Calibration of Accelerometer Output for Children. Med Sci Sports Exerc. 2005 Nov;37(11 Suppl):S523-30. doi:10.1249/01.mss.0000185658.28284.ba.: page S529 “Behavioral approaches (e.g., direct observation) for calibrating accelerometry output may be particularly useful when studying young children where measurement and interpretation of energy expenditure data are difficult tasks.”

5. Line 61, page 2: Why classifying into mobile and stationary? Maybe you can add one sentence to clarify this already in the abstract.

+++ This classification goes back to the terminology consensus project process and outcome (Ref. 65,Tremblay et al, 2017). ‘Stationary’ is defined as the superordinate concept uniting different types of behaviours: sedentary (sitting), standing, screen time, reclining, lying. ‘Mobile’ is the superordinate concept uniting light PA, moderate PA and vigorous PA.

6. Line 65, page 2: I wonder what you mean with context information. Please specify.

+++ Contextual information of behaviour refers to the knowledge of what the children were doing over time while measurements took place. Movement intensities of the children’s real habitual repertoire - and not arbitrary exercise, like dancing to a specific beat – is less prone to under- and overestimation of time spent in a given intensity range in observational studies.

+++ We amended the abstract, saying: … Receiver-Operating-Curve classification was applied to determine activity thresholds and to assign two composite movement classes.

For point 5 and 6, we would have added the reference of Tremblay et al (2017) in the abstract, but references are not allowed in the abstract.

Introduction

General:

- Throughout the whole introduction, sentences are very long. I would recommend, separating long sentences (especially Line 78-84, page 3; Line 129-136, page 5).

+++ Revised and shorter.

- The introduction is very long. I would recommend shorten it. The history of accelerometers is not that important in my opinion (Line 73, page 3). Further, some information could be more summarized. Please critically check the introduction for shortening.

1. Line 83, page 3: World Health Organization (WHO)

+++ Included.

2. Line 140, page 5: What is the intention to assess context-related movements and distinct between mobile and stationary? From a public health perspective, it is necessary and more common to focus on intensities, as especially MVPA leads to health benefits and LPA does not impact health status in that way.

+++ Context-related movements refer to movements that are required for different behaviours. It is outlined in the Terminology Consensus Project by Tremblay et al (2017). Stationary means you are awake and you do something, where you can move hands, arms, legs and body, but without changing position, which relates to body postures described as standing, squatting, lying, reclining, but not sleeping.

Mobile includes moving your body to a different position, which can include LPA as well as MVPA. Our study primarily reports useful cut-off points to identify MVPA and one of our aims was to differentiate between different intensities of PA.

When you have small children, the level of movements is mechanically and anatomically different from adult PA. While the line is drawn between MVPA and LPA in adults, this does not necessarily work for small children. The intention is to draw the line between stationary activities and LMVPA. Playing on the floor, in the sandpit, in the snow or climbing trees, which adolescents and adults hardly do, belong to LPA and should not be underestimated for its health benefits for small children for sensory-muscular-eye coordination, balance and cognitive development. Reilly et al (2003) has pointed the way to not only measure physical activity levels but also (stationary) inactivity duration for its usefulness as an outcome measure for premature adiposity and/or risk factors, see OBESITY RESEARCH Vol. 11 No. 10 October 2003.

Method

1. Page 6, Line 150: Healthy 3-year-old children. What does that mean? Without disabilities? Without illness? I would recommend rephrasing the sentence in a Subject-first language (3-year old children without…).

+++ Done

It says now: 3-year-old children without any chronic disease or weight outside the normative range (± 2 standard deviations) using a Swedish growth reference [51] were included.”

2. Page 7, Line 170: Did you use expected MET-values? How did the classification of the activities to the different intensities took place?

+++ We used direct observations during the pilot in characterising movement patterns into intensities from our expertise within the group of 27 years of analysing actigraphic patterns, from newborn babies to the elderly. Specifically, we came up with a list of behaviours that demand different intensities and tested them out in the pilot. Once we had behaviours, which the children liked doing, we ranked them to become progressively more intense.

3. Page 7, Line 181: On which basis did you chose the activities?

+++ By expertise in child behaviour and actigraphic time series analysis.

4. Page 8, Line 197: I wonder why you didn’t validate your data with the heart rate when you have the data anyway? Did you do something to validate the data? Or just the correlation between two similar devices?

+++ The criterion here is direct observation in calibrating two different models at two different positions for six different behaviours in a narrow age range. The heart rate data will get their own space in a separate manuscript and refer to this study.

The devices measure what they are intended to measure, but given their different sampling frequencies and sensors, the different acceleration from wrist and waist movements will likely differ in counts and therefore need to be compared within and between devices under observed behaviours The correlations between the actigraphic outputs for the two pairs of devices provide a level of consistency.

5. Page 8, Line 211. Please remove the space between 250 g.

+++ Done.

6. Page 9, Line 219: The raw data of the accelerometer are accelerations (g as unit). How do you get counts from the acceleration?

+++ via the piezoelectric effect, where the acceleration in the piezoelectric sensing material is turned into an electrical charge, and the level of charge is then converted into a count, which is proportional to the level of acceleration, see piezoelectric sensor (https://en.wikipedia.org/wiki/Piezoelectric_sensor).

7. Page 9, Line 225: proprietary software – what is this? I am not sure if everybody know what this means. Please contextualize.

+++ replaced by Motionware software

8. Page 9, Line 226: what is the behavioral observation protocol. Please describe this in more detail. This term is frequently in your manuscript but for me it is not sure what is meant with that.

+++ please find above under point 4 .

9. Page 9, Line 231: Why didn’t you use test-retest to assess the reliability? How did you assess the reliability? Known studies often focus on validity and reliability in the context of calibration.

+++ We calibrated behavioural classes using movement intensities. Reliability refers to the consistency of a measure. A high test-retest correlation makes sense when the construct being measured is assumed to be consistent over time, which is the case for example for intelligence, but constructs of behaviour under the influence of emotions (how they feel right now) are not assumed to be stable over time. The very nature of behaviour that produces a low test-retest correlation over a period of a week or so would not be a cause for concern. Validity is the extent to which the intensity from a measurement represents the behaviour it intends to cover. We use content validity as the extend to which the measurement method appears “to cover” the construct of behavioural class of interest. For this we use the device models for quantitative assessment and checking the measurement against the conceptualised definition by the ‘Direct observation protocol’. We used ‘direct observation’ as criterion.

10. Page 10, Line 241: observer-based behaviors – what is this? Please contextualize.

+++ please find above under point 4 and figure 1, Bottom explanatory table.

11. Page 10, Line 246: I wonder why you make a distinction between mobility and stationary behavior when the common classification are the intensity levels.

+++ We use classification guided by behaviour not intensity levels. Behaviour-based movement cut-offs means movement cut-offs derived from intensity levels. We do not use intensity classifications for physical activity only – we look holistically into behaviours.

12. Page 10, Line 252: on which basis did you do that? MET-values?

+++ We did not use exertion of energy expenditure during physical activity in the children. We used behaviour, broadly defined as anything a living being does, which includes actions. Here, we used the actions of different behaviours as described.

Results

1. Do you have the individual raw data/activity counts of each conducted activity?

+++ We do have the activity counts/30s as raw data over each person and conducted behaviour and have provided open access to non-commercial use.

2. Page 16, Table 3: Cut-off values per which second?

+++ We used counts per 30s.

Discussion

1. General: Some paragraphs are difficult to understand. Maybe it would be helpful to add some subheadings to help the reader to go through the discussion (e.g., cut-off values, sensor position, …) see also (Beck et al., 2023).

+++ We have included sub-headings into the discussion.

2. Page 16, Line 382: Again, what is the observation technique? Are there no results concerning this?

+++ I refer to #4 above on the technique. Regarding results of this technique, we used the start and end times to identify and extract each sequence of behaviour correctly in the time series. Observation was the choice over indirect calorimetry because of the age of the children.

The observation results are reflected in the cumulated wear-time of devices by behaviour, reported in the supplementary figure S1, Same duration for every behaviour avoids introducing a bias from unequal variation from different durations of the behaviours performed.

3. Page 16, Line 383: Children’s activity rating scale  in your study? The sentence is not clear to me.

+++ We have amended the sentences to clarify.

4. Page 19, 390 ff: Is this also a phenomenon found in actual literature as you mention “Already two decades ago”?

+++ . ‘Already’ refers to the fact that this method has been used in a calibration study in healthy 3-5 year old children. To our knowledge, indirect calorimetry has been used but only in older children (Freedson et al, 2005) or

---

## [Decision Letter · Decision Letter 1]

17 Dec 2024

Behaviour-based movement cut-off points in 3-year old children comparing wrist- with hip-worn actigraphs MW8 and GT3X

PONE-D-24-07355R1

Dear Dr. Wulff,

We’re pleased to inform you that your manuscript has been judged scientifically suitable for publication and will be formally accepted for publication once it meets all outstanding technical requirements.

Kind regards,

Duncan S Buchan

Academic Editor

PLOS ONE

Additional Editor Comments (optional):

Reviewers' comments:

Reviewer's Responses to Questions

**Comments to the Author**

1. If the authors have adequately addressed your comments raised in a previous round of review and you feel that this manuscript is now acceptable for publication, you may indicate that here to bypass the “Comments to the Author” section, enter your conflict of interest statement in the “Confidential to Editor” section, and submit your "Accept" recommendation.

Reviewer #1: All comments have been addressed

Reviewer #2: All comments have been addressed

2. Is the manuscript technically sound, and do the data support the conclusions?

Reviewer #1: Yes

Reviewer #2: Yes

3. Has the statistical analysis been performed appropriately and rigorously? 

Reviewer #1: Yes

Reviewer #2: Yes

4. Have the authors made all data underlying the findings in their manuscript fully available?

Reviewer #1: Yes

Reviewer #2: Yes

5. Is the manuscript presented in an intelligible fashion and written in standard English?

Reviewer #1: Yes

Reviewer #2: Yes

6. Review Comments to the Author

Reviewer #1: Dear Authors,

Thank you for the thorough revision of your manuscript. The revisions were carried out with great care, and all comments and suggestions have been addressed in an excellent manner.

Based on the significant improvements and the high quality of the work, I am pleased to recommend your manuscript for acceptance.

I sincerely appreciate your efforts and wish you continued success in your research.

Reviewer #2: A thorough revision of an already excellent paper. Thank you for the nice read. No further recommendations!

7. PLOS authors have the option to publish the peer review history of their article (what does this mean? ). If published, this will include your full peer review and any attached files.

**Do you want your identity to be public for this peer review?** For information about this choice, including consent withdrawal, please see our Privacy Policy .

Reviewer #1: No

Reviewer #2: No

---

## [Editor Report · Acceptance letter]

PONE-D-24-07355R1

PLOS ONE

Dear Dr. Wulff,

I'm pleased to inform you that your manuscript has been deemed suitable for publication in PLOS ONE. Congratulations! Your manuscript is now being handed over to our production team.

Kind regards,

on behalf of

Dr. Duncan S Buchan

Academic Editor

PLOS ONE